# Assessment of Genetic Diversity of the “Acquaviva Red Onion” (*Allium cepa* L.) Apulian Landrace

**DOI:** 10.3390/plants9020260

**Published:** 2020-02-18

**Authors:** Luigi Ricciardi, Rosa Mazzeo, Angelo Raffaele Marcotrigiano, Guglielmo Rainaldi, Paolo Iovieno, Vito Zonno, Stefano Pavan, Concetta Lotti

**Affiliations:** 1Department of Soil, Plant and Food Sciences, Plant Genetics and Breeding Unit University of Bari, Via Amendola 165/A, 70125 Bari, Italy; luigi.ricciardi@uniba.it (L.R.); angelo.marcotrigiano@uniba.it (A.R.M.); vito.zonno@uniba.it (V.Z.); stefano.pavan@uniba.it (S.P.); 2Department of the Sciences of Agriculture, Food and Environment, University of Foggia, Via Napoli 25, 71122 Foggia, Italy; 3Department of Biosciences, Biotechnologies and Biopharmaceuticals, University of Bari, Via Orabona 4, 70125 Bari, Italy; guglielmo.rainaldi@uniba.it; 4Department of Energy Technologies, Bioenergy, Biorefinery and Green Chemistry Division, ENEA Trisaia Research Center, S.S. 106 Ionica, km 419+500, 75026 Rotondella (MT), Italy; paolo.iovieno@enea.it

**Keywords:** SSR markers, onion landraces, genetic diversity, morphological and quality traits

## Abstract

Onion (*Allium cepa* L.) is the second most important vegetable crop worldwide and is widely appreciated for its health benefits. Despite its significant economic importance and its value as functional food, onion has been poorly investigated with respect to its genetic diversity. Herein, we surveyed the genetic variation in the “Acquaviva red onion” (ARO), a landrace with a century-old history of cultivation in a small town in the province of Bari (Apulia, Southern of Italy). A set of 11 microsatellite markers were used to explore the genetic variation in a germplasm collection consisting of 13 ARO populations and three common commercial types. Analyses of genetic structure with parametric and non-parametric methods highlighted that the ARO represents a well-defined gene pool, clearly distinct from the Tropea and Montoro landraces with which it is often mistaken. In order to provide a description of bulbs, usually used for fresh consumption, soluble solid content and pungency were evaluated, showing higher sweetness in the ARO with respect to the two above mentioned landraces. Overall, the present study is useful for the future valorization of the ARO, which could be promoted through quality labels which could contribute to limit commercial frauds and improve the income of smallholders.

## 1. Introduction

The *Allium* genus includes about 750 species [1], among which onion (*Allium cepa* L., 2n = 2x = 16) is one of the most widespread. *A. cepa* has a biennial cycle and outcrossing reproductive behavior. Nowadays, onion global production (97.9 Mt) makes it the second most important vegetable crop after tomato [2]. Since olden times, onion bulbs have been used both as food and in folk medicinal applications. Indeed, ancient Egyptians already reported several therapeutic formulas based on the use of garlic and onions in a medical papyrus of the 1550 BC, the Codex Ebers [3].

This versatile and healthy vegetable is consumed raw, fresh, or as processed product, and used to enhance the taste of many dishes. Several recent studies claim that onion consumption may reduce risk of cardiovascular diseases [4,5], obesity [6], diabetes [7], and various forms of cancer [8,9,10]. Onion health proprieties are often attributed to high levels of two classes of nutraceutical compounds: flavonoids and alk(en)yl cysteine sulphoxides (ACSOs). The first class includes flavonols and anthocyanins. Quercetin is the main detectable flavonol, known for its strong antioxidant and anti-inflammatory properties in free radical scavenging and transition metal ions binding [11]; whereas anthocyanins confer red/purple color to some onion varieties. As for ACSOs, the most abundant is isoalliin [(+)-trans-S-1-propenyl-L-cysteine sulfoxide] [12], a non-volatile and non-proteinogenic sulfur amino acid stored in the cells, which is indirectly responsible for the pungent aroma and taste of onions [13]. Upon tissue disruption, isoalliin is cleaved by the enzyme alliinase to produce a series of volatile compounds (pyruvate, ammonia, thiosulphonates and propanethial S-oxide) which induce tearing and cause unpleasant smell (pungency) [14]. The onion pungency is often measured as the amount, per gram of fresh weight, of pyruvic acid generated by hydrolysis [15,16].

In the countries of the Mediterranean basin, proposed as one of the secondary diversity centers of *A. cepa* [17,18], onion bulbs display a wide variability in shape, size, color, dry matter, and pungency [19,20,21,22]. Moreover, sulphur-based fertilization, agronomic practices, type of soil, climatic conditions, and the genotype of cultivars or landraces can influence the bulb quality by conferring peculiar organoleptic and nutritional values [23,24,25,26,27]. In Italy, despite the wide onion germplasm availability, only a few onion varieties are often subjected to scientific studies and properly characterized [28,29].

Thoroughly genetic and phenotypic characterization of agro-biodiversity is crucial to assure appropriate conservation of plant genetic resources and promote the use of specific genotypes in the value chain [30,31,32]. Simple sequence repeat (SSR) markers have been often chosen for mapping [33,34,35], DNA fingerprinting and cultivar discrimination [36,37,38], and reliable estimation of genetic variability within and among landraces [39,40,41,42], since they are locus specific, multi-allelic, codominantly inherited, highly reproducible, and suitable for automated genotyping.

In the present study, we focused our attention on an Apulian traditional landrace, the “Acquaviva red onion” (ARO), which is cultivated according to organic farming methods in a small area of the town of Acquaviva delle Fonti, in the province of Bari (Apulia, Southern Italy). The bulbs of this landrace are big and flattened and red colored and are largely used in local recipes. Although the ARO gained the “Slow Food Presidium” quality mark, its production could be further promoted and protected by European Union quality marks such as protected geographical indication (PGI) and protected designation of origin (POD), as these might contribute to limit the commercial frauds and improve the income of smallholders. Herein, SSR molecular markers were used as powerful tools to assess genetic variation among ARO populations and to discriminate this landrace from other two Southern Italian red onion landraces. Furthermore, we estimated pungency and soluble solid content in order to evaluate ARO flavor in relation with the market demand.

## 2. Results

### 2.1. Establishment of Acquaviva Red Onion Germplasm Collection and Morphological Characterization

Seeds of 13 populations of the ARO landrace, donated by farmers in the framework of the BiodiverSO Apulia Region project were used to establish an ARO germplasm collection.

Morphological descriptors, related to bulb, skin, and flesh were collected on ARO germplasm and on three onion landraces, two belonging to the “Tropea red onion” (TRO) landrace and one to the “Montoro copper onion” (MCO) landrace (Figure 1). All the ARO bulbs were flat and were characterized by red external skin and flesh with different shades of red. In contrast, the flesh of TRO bulbs were fully red, whereas the flesh of MCO bulbs was poorly pigmented (Appendix A). Biochemical analysis allowed to evaluate the solid soluble content and pungency. As reported in Table 1, the mean values of solid soluble content of bulbs in ARO populations was 7.60, and ranged from 6.00 (ARO12) to 9.50° Brix (ARO11 and ARO13). This value was higher than the one estimated for the TRO and MCO landraces (4.25 and 6.00° Brix, respectively).

The mean value of ARO pungency, assessed by means of pyruvic acid content, was 6.00, ranged from 4.51 µmol g^−1^ FW (ARO6) to 7.04 (ARO8). This value was higher than the one estimated in TRO and MCO landraces (3.54 µmol g^−1^ FW and 4.18 µmol g^−1^ FW, respectively).

### 2.2. SSR Polymorphism and Genetic Relationships among Accessions

In the present study, 11 out of 37 tested SSR primer combinations provided single-locus polymorphisms, i.e., yielding at most two amplification products in a single individual. Overall, 55 alleles were detected in 320 individuals with a number of alleles per locus ranging from 2 (ACM147 and ACM 504) to 11 (ACM132) and a mean value of 5 alleles (Table 2). In individual populations, the number of alleles (Na) ranged from 1.94 (ACM147 and ACM504) to 5.38 (ACM132), whereas the effective number of alleles (Ne) ranged from 1.41 (ACM152) to 2.82 (ACM449). Discrepancies between Na and Ne values were due to the presence of alleles with low frequency in the populations and the predominance of only a few alleles. The highest observed heterozygosity (Ho) value was highlighted for ACM138 and ACM449 (0.62), whereas the lowest one was associated with ACM152 (0.25). Expected heterozygosity (He), which corresponds to the theoretical expectation in a panmictic population, ranged from 0.37 (ACM504) to 0.61 (ACM132, ACM138, and ACM449). The Wright’s fixation index (F_IS_), displayed values close to zero (average 0.05) for all the markers, indicating similar values between observed and expected heterozygosity levels, as expected for an outcrossing species. The efficiency of individual SSR marker in genetic fingerprinting was estimated by the polymorphic information content (PIC) index, with a mean value of 0.48 and ranged from 0.33 (ACM504) to 0.67 (ACM132). Another efficiency index, the Shannon’s Information Index (I) displayed a mean value of 0.84, and assumed values ranged from 0.45 (ACM152) to 1.20 (ACM132).

### 2.3. Analysis of Molecular Variance and Genetic Structure

Hierarchical partitioning of genetic variation among and within populations was computed by AMOVA. The results highlighted a considerable fraction of genetic variation within populations (87%). Variation among populations, 13%, was highly significant (*P* < 0.001) (Table 3). Pairwise values of the Φ_PT_ parameter, an analogous of the Wright’s F_ST_ fixation index, ranging from 0.002 (ARO2/ARO10) to 0.468 (ARO7/TRO2), were significant (*P* < 0.05), except for nine pairwise comparisons (Appendix A).

Investigation of genetic structure in the *A. cepa* collection genotyped in this study was performed by means of the admixture model-based clustering analysis implemented in the software STRUCTURE. The Evanno ΔK method suggested subdivision in two clusters (K = 2) as the most informative for our dataset, with the next highest peak at K = 5 (Appendix A). As for K = 2, all populations were assigned to one of the two clusters with a membership coefficient (q) ≥ 0.7. As shown in Figure 2a, the first cluster (named S1) included MCO and all ARO populations, whereas the S2 cluster grouped the two TRO populations. At K = 5, providing a deeper description of the dataset (Figure 2b), 75% of the accessions were assigned to one of the five cluster. Separation between ARO (S1) and TRO (S2) was confirmed, although some ARO populations were admixed (q < 0.7) or grouped separately in the two new clusters S3 and S4 (ARO7 and ARO12, respectively). Interestingly, the MCO commercial type formed a distinct cluster (S5) separated from the Apulian red onion.

### 2.4. Genetic Relationships among Populations

SSR polymorphism allowed to draw a dendrogram of genetic diversity and the results of the phylogenetic analysis are shown in Figure 3a. Here, the germplasm collection was splitted in five groups strongly supported by bootstrap values. The ARO7 and ARO12 populations were immediately separated from the remaining populations and formed two distinct clusters. The third cluster included the two commercial populations of TRO, meanwhile the fourth node divided MCO from eleven ARO populations. Genetic relationship occurring among populations was further investigated by means of principal coordinate analysis (PCoA) (Figure 3b). As previously highlighted, ARO populations were grouped tightly, except for ARO12 and ARO7, which appeared in isolated positions in the PCoA plot. The two TROs and the MCO populations were scattered in the lower-right panel of the plot.

## 3. Discussion

Within the large amount of agro-biodiversity traditionally cultivated in the Southern Italy, onion landraces represent niche products that need to be preserved from the risk of genetic erosion and the threat of replacement by modern cultivars. In the framework of the regional project BiodiverSO, aimed at collecting, characterizing, promoting, and safeguarding genetic resources of the Apulia region strongly linked to local heritage, we established a seed collection of 13 populations of the ARO landrace. We reported the first assessment of ARO variation in terms of DNA polymorphisms and two biochemical parameters, soluble solid and pyruvic acid contents, related to flavor traits and of importance for the acceptance of the fresh uncooked products. In addition, data on the ARO landrace were compared with those collected on two other pigmented onion landraces with which it often mistaken.

Biochemical analyses highlighted the sweetness of the 13 ARO populations, related to high soluble solid content and medium pungency, according to the sweet onion industry guidelines [31]. ARO bulbs were sweeter than those of the TRO and MCO landraces, and displayed a slightly higher pungency. However, sweetness in onions is due to a balance between sugar content and pungency, therefore this characterization could be useful to support the selection of genotypes of value, usually carried out by farmers only based on the morphology.

SSR markers were confirmed to be a useful tool to discriminate genotypes, albeit collected within a narrow growing area such as the town of Acquaviva delle Fonti. The selected markers displayed higher number of alleles than the markers previously reported by [43] and [44], but lower than the markers reported by [45]. Moreover, 50% of our set of markers showed PIC index values greater than 0.5, proving to be suitable to discriminate the populations in the collection, as suggested by [46]. Assessment of diversity within populations revealed similar values between Ho and He, resulting in low F_is_ values. This is in agreement with the out-crossing nature of *A. cepa,* which seriously suffers from inbreeding depression [47]. The overall F_is_ value calculated in onion populations considered in this study (0.054) was lower than that the one previously reported by [45] (0.22) and almost identical to the one found by [31] (0.08) and [48] (0.00) who assessed genetic diversity in onion landraces from northwest Spain and Niger, respectively. Noteworthy levels of heterozygosity in ARO populations reinforce the notion that Apulia represents a diversity center for many horticultural species [32,42,49,50,51].

AMOVA highlighted that most molecular variation in the collection genotyped in this study lies within the populations. However, significant genetic differentiation among populations (Φ_PT_ values) revealed the occurrence of genetic stratification. In fact, although our results indicated the presence of genetic uniformity in most ARO populations, forming a well-defined cluster, the ARO7 and ARO12 populations displayed a clearly distinct genetic profile. This result could be due to a different origin of seeds used by the two farmers from which the populations were collected. Moreover, based on the results obtained, the ARO landrace can be considered clearly distinct at the genetic level from the TRO and MCO landraces. In a recent study, [29] assessed the genetic diversity of several Italian onion landraces including “Acquaviva,” “Tropea,” and “Montoro.” Although the authors used SNP markers to assess the genetic diversity of a wider onion collection, the genotyping was not able to discriminate “Acquaviva” from “Tropea” and “Montoro” onions. Probably, this discrepancy is due to the low mean PIC value found (0.292), suggesting a modest general informativeness of the loci under analysis as claimed by [29]. Furthermore, in order to investigate the presence of sub-structure in their Italian cluster, it would have been better to analyze the Italian genotypes separately from the rest of the collection. Probably it would have allowed to visualize pattern of genetic diversity linked to geographic stratification or traits under empiric selection.

In conclusion, the present study represents a comprehensive report on an onion landrace associated with local cultural heritage and of economic importance for the farmers. Our results highlight that, with a few exceptions, ARO is characterized by a well-defined gene pool, which deserves to be preserved from the risk of genetic erosion. Therefore, the establishment of a representative collection of this valuable source of genetic diversity has been crucial. Finally, the genetic and phenotypic characterization of ARO might be useful to obtain quality marks from the European Union.

## 4. Materials and Methods

### 4.1. Germplasm Collection, Plant Material, and DNA Extraction

A set of 13 populations of the ARO landrace were acquired within the framework of an Apulia Region project (BiodiverSO: https://www.biodiversitapuglia.it/), through a series of missions carried out in “Acquaviva delle Fonti”, a small Apulian town in Province of Bari, Italy. Collection sites of each accessions were mapped through the Geographic Information System (GIS) and reported in Table 4. In addition, two populations from the TRO landrace and one population from the MCO landrace were included in the present study and used as references. All the plant material was grown in the same environmental conditions at the experimental farm “P. Martucci” of the University of Bari (41°1′22.08″ N, 16°54′25.95″ E), under protection cage to avoid cross pollination among populations and assuring intra-population pollination by means of blowflies (*Lucilia caesar*). The 16 populations were characterized for traits related to bulb size and shape and skin and flesh color (Appendix A). In addition, solid soluble content assay was performed using a hand-held refractometer and pungency was measured in onion juice samples adding 2,4-dinitrophenyl hydrazine (0.125% *v/v* in 2N of HCl) and evaluating absorbance at 420 nm, as reported by [31]. The Duncan’s multiple-range test and the SNK test were carried out to determine the presence of significant differences.

Leaf material of 20 genotypes per population were sampled and stored at −80 °C until use. For polysaccharide-rich species, as *A. cepa*, first steps removing polysaccharide are essential to obtain good-quality DNA, therefore initial washes in STE buffer (0.25 M sucrose, 0.03 M Tris, 0.05 M EDTA) were performed as described by [52]. Total DNA was extracted following the CTAB method [53] and finally it was checked for quality and concentration by Nano Drop 2000 UV–vis spectrophotometer (ThermoScientific, Waltham, MA, USA) and 0.8% agarose gel electrophoresis.

### 4.2. SSR Analysis

16 EST-SSR primer combinations developed by [54] and previously tested in genetic diversity studies by [43] and [44] and 21 genomic SSR [45,46,47,48,49,50,51,52,53,54,55] were screened to evaluate their suitability (Appendix A). Genotyping was performed using the economic fluorescent tagging method in which the M13 tail is added to each forward SSR primer [56]. PCR mixes were prepared in 20 µL reaction containing: 50 ng of total DNA, 0.2 mM of dNTP mix, 1X of PCR reaction buffer, 0.8 U of DreamTaq DNA polymerase (Thermo Scientific, Waltham, MA, USA), 0.16 µM of reverse primer, 0.032 µM of forward primer extended with the M13 sequence (5′-TGTAAAACGACGGCCAGT-3′), and 0.08 µM of a universal M13 primer labelled with FAM or NED fluorescent dyes (Sigma-Aldrich, St. Louis, MO, USA). The PCR reactions were carried out in the SimpliAmp (Applied Biosystems, CA, USA) thermocycler with the following conditions for the majority of the primer pairs: 94 °C for 5 min, 40 cycles at 94 °C for 30 s, 58 °C for 45 s and 72 °C for 45 s and a final elongation at 72 °C for 5 min. As for ACM446 and ACM449, a touchdown PCR was applied with annealing of 60 °C to 55 °C over 10 cycles, 30 cycles at 55 °C, followed by a final extension of 5 min at 72 °C. PCR products were loaded into a 96-well plate and mixed with 14 µL of Hi-Di Formamide (Life Technologies, Carlsbad, CA, USA) and 0.5 µL GeneScan 500 ROX Size Standard (Life Technologies, Carlsbad, CA, USA). Amplicons were resolved by means of ABI PRISM 3100 Avant Genetic Analyzer (Life Technologies, Carlsbad, CA, USA) capillary sequencing machine, where the alleles were scored as co-dominant and assigned by using the GeneMapper Software Version 3.7.

The softwares GenAlEx 6.5 [57] and Cervus 3.0.7 [58] were used to estimate number of alleles (Na), number of effective alleles (Ne), observed heterozygosity (Ho), expected heterozygosity (He), polymorphic information content (PIC), Shannon’s information index (I), and fixation index (Fis) for each SSR locus.

### 4.3. Assessment of Genetic Diversity

Hierarchical partitioning of genetic variation among and within the onion populations was evaluated by GenAlEx 6.5 [57] through the analysis of molecular variance (AMOVA) with 999 bootstrapping to test for significance. Moreover, GenAlEx 6.5 software was used to estimate the diversity within each population by computing the average of Ho, He, and Fis over all the SSR loci.

Population structure was inferred by the Bayesian model-based clustering algorithm implemented in the STRUCTURE v.2.3.4 software [59]. The data set was run with a number of hypothetical clusters (K), ranging from 1 to 10, setting ten independent runs per each K value. For each run, aiming to verify the consistence of results, 100,000 initial burn-in period and 100,000 Markov Chain Monte Carlo (MCMC) iterations were performed under the admixture model and independent allele frequencies among populations. The most likely K value was determined implementing the ΔK method, described by [60], in the web-based program STRUCTURE HARVESTER [61]. An individual population was assigned to a specific cluster when its membership coefficient (q-value) was higher than 0.7, otherwise it was considered of admixed ancestry.

Principal coordinate analysis was performed in order to visualize patterns of genetic relationship among accessions revealed by the Nei’s genetic distance matrix (Appendix A). Based on allele frequencies, a dendrogram of genetic distance was constructed implementing the unweighted pair group method with arithmetic averages (UPGMA) cluster analysis in the POPTREEW software [62]. Bootstrapping was applied to assess the confidence in hierarchical clustering, setting 100 resampling of the data set. Finally, MEGA X software [63] was used as tree drawing software.

## Figures and Tables

**Figure 1 plants-09-00260-f001:**
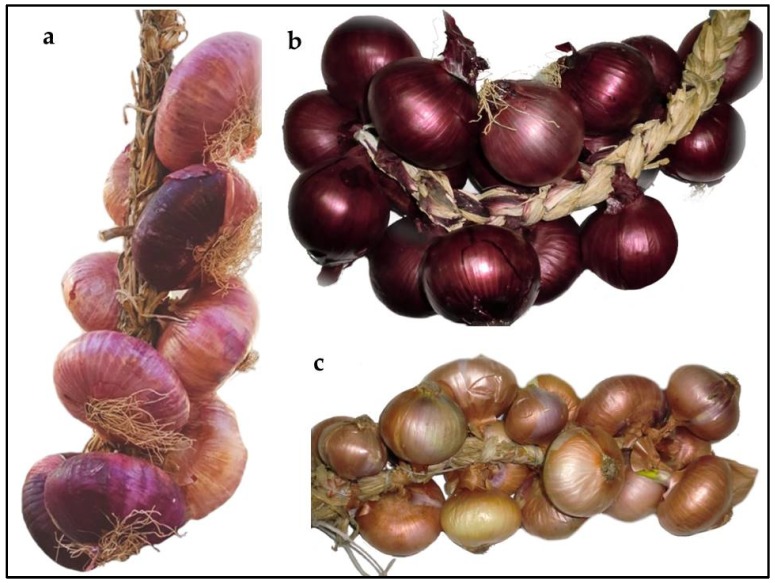
Bulbs of “Acquaviva red onion” (**a**) “Tropea red onion” (**b**) and “Montoro copper onion” (**c**) landraces.

**Figure 2 plants-09-00260-f002:**
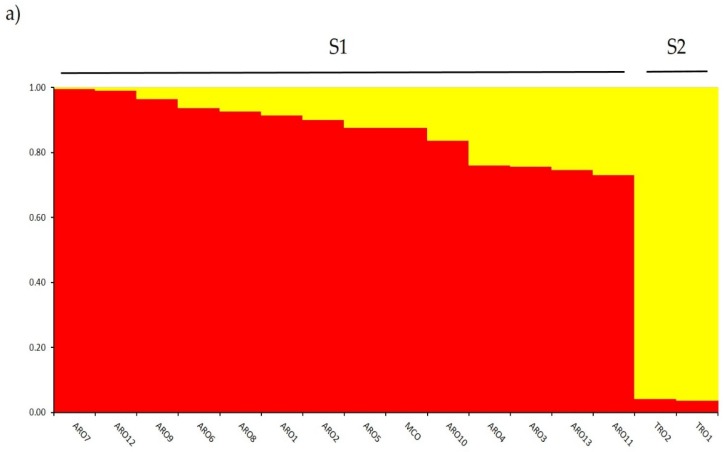
Estimated structure of the *Allium cepa* germplasm collection characterized in this study. Results are shown for K = 2 (**a**) and K = 5 (**b**). The *y*-axis indicates the estimate membership coefficient (q). Each of the 16 onion populations is represented by a single vertical bar, which is partitioned into colored segments in proportion to the estimated membership coefficient in each of 2 (**a**) and 5 (**b**) clusters.

**Figure 3 plants-09-00260-f003:**
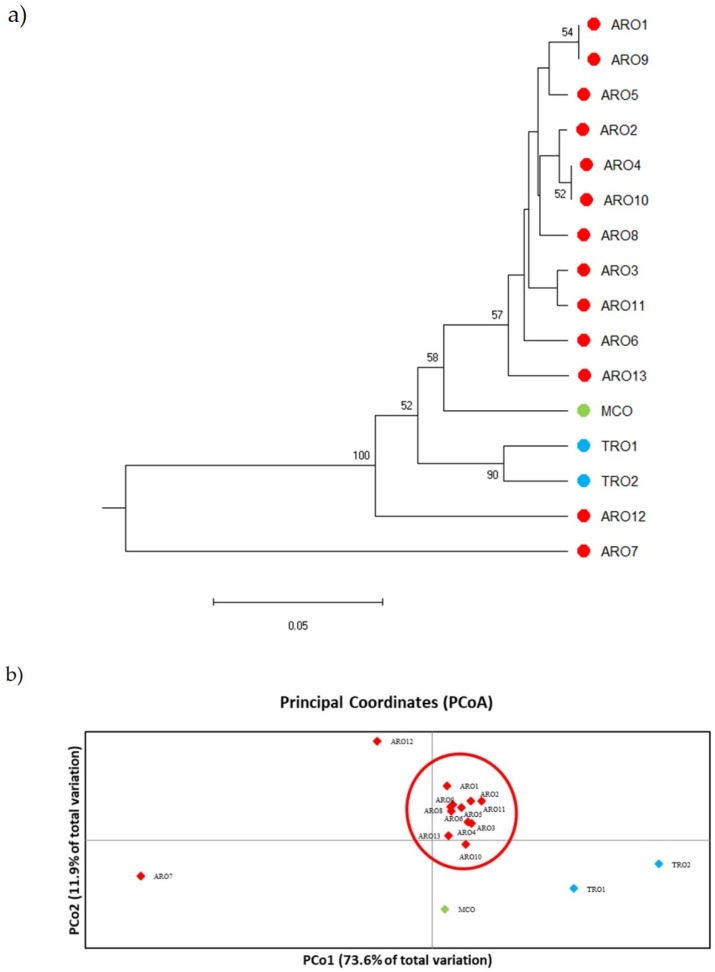
Genetic diversity among 16 *A. cepa* populations characterized in this study, based on their SSR profile. (**a**) UPGMA dendrogram of genetic distance. Bootstrap support values ≥50 are indicated above the corresponding nodes; (**b**) principal component analysis (PCoA). The cluster circled in red fully matched with the group generated by phylogenetic analysis and constituted by 11 ARO accessions.

**Table 1 plants-09-00260-t001:** Solid Soluble Content and Pungency Values Assessed in “Acquaviva Red Onion” (ARO), “Tropea Red Onion” (TRO), and “Montoro Copper Onion” (MCO) Populations *.

	Soluble Solid Content (Brix)		Pungency (µmol g^−1^ FW)	
CODE	Mean	CV ^†^ (%)	Mean	CV ^†^ (%)
ARO1	6.25 D *	5.65	5.84 ab *	23.78
ARO2	7.25 DC	4.87	6.51 a	22.98
ARO3	7.50 BCD	9.42	5.28 ab	22.88
ARO4	7.50 BCD	0.00	6.97 a	3.74
ARO 5	7.50 BCD	0.00	6.80 a	9.68
ARO6	6.25 D	5.65	4.51 ab	39.18
ARO7	7.25 DC	4.87	5.25 ab	15.44
ARO8	9.00 AB	0.00	7.04 a	3.49
ARO9	8.25 ABC	4.28	6.84 a	0.15
ARO10	7.00 DC	0.00	5.94 ab	6.57
ARO11	9.50 A	7.44	5.54 ab	16.43
ARO12	6.00 D	0.00	4.91 ab	9.70
ARO13	9.50 A	7.44	6.63 a	24.93
MCO	6.00 D	0.00	4.18 ab	2.66
TRO1	4.25 E	8.31	2.80 b	2.10
TRO2	4.25 E	8.31	4.28 ab	4.79

* Means with the same letters in uppercase or lowercase are not statistically different at 0.01P or 0.05P, respectively (SNK’s Test). ^†^ Coefficient of variation.

**Table 2 plants-09-00260-t002:** Polymorphism Features of the 11 SSR Markers Used to Estimate Genetic Diversity in ARO, TRO, and MCO Populations. Total Number of Alleles (Na), Band Size Range, and Polymorphic Information Content (PIC) Index Refer to the Total Set of 320 Individuals Genotyped in this Study. Number of Alleles (Na), number of Effective Alleles (Ne), Observed Heterozygosity (Ho), Expected Heterozygosity (He), Fixation Index (F_is_), and Shannon’s Information Index (I) refer to Mean Values Calculated from 16 Populations, Each Composed by 20 Individuals.

Locus.	Total Na	Size Range (bp)	PIC	Mean					
				Na	Ne	Ho	He	I	F_is_
**ACM91**	4	189–205	0.40	2.63	1.72	0.38	0.39	0.66	0.04
**ACM101**	4	229–241	0.52	2.94	2.37	0.53	0.56	0.92	0.06
**ACM132**	11	186–248	0.67	5.38	2.78	0.55	0.61	1.20	0.09
**ACM138**	5	242–272	0.66	3.69	2.82	0.62	0.61	1.09	−0.02
**ACM147**	2	264–266	0.37	1.94	1.83	0.44	0.44	0.62	−0.01
**ACM152**	4	228–244	0.25	2.38	1.41	0.25	0.27	0.45	0.07
**ACM235**	4	286–298	0.41	2.81	1.77	0.44	0.41	0.72	−0.06
**ACM446**	6	108–120	0.56	3.50	2.48	0.49	0.58	1.01	0.16
**ACM449**	8	120–140	0.66	4.88	2.82	0.62	0.61	1.18	−0.03
**ACM463**	5	202–210	0.47	3.38	1.95	0.46	0.48	0.83	0.05
**ACM504**	2	188–192	0.33	1.94	1.64	0.30	0.37	0.54	0.20
**Mean**	5		0.48	3.22	2.15	0.46	0.48	0.84	0.05

Among the populations, ARO3, ARO6, ARO8, ARO10, TRO1, and MCO displayed high level of genetic variation (Ho ≥ 0.5), whereas the lowest diversity was observed in the population ARO7 (Ho = 0.27) (Appendix A). Overall, all the accessions displayed F_is_ values close to zero (F_is_ mean value = 0.054), as expected under random mating conditions.

**Table 3 plants-09-00260-t003:** Analysis of Molecular Variance of 320 Genotypes from 16 Populations of *Allium cepa* L.

Source	df	Sum of Squares	Variance Estimation	Variance (%)	Φ_PT_	*P*
**Among populations**	15	458.63	1.16	13%	0.134	0.001
**Within populations**	304	2272.99	7.50	87%		
**Total**	319	2731.62	8.66			

**Table 4 plants-09-00260-t004:** List of Populations Collected and Genotyped in This Study. For Each Population, Identification Code, Local Name, GPS Coordinate, and Gene Bank Preserving the Seeds are Reported.

Code	Name	GPS Coordinates	Gene Bank ^†^
ARO1	Cipolla rossa di Acquaviva	40°54′21.708″ N 16°49′1.631″ E	Di.S.S.P.A.
ARO2	Cipolla rossa di Acquaviva	40°53′14.28″ N 16°48′56.879″ E	Di.S.S.P.A.
ARO3	Cipolla rossa di Acquaviva	40°54′11.304″ N 16°49′13.079″ E	Di.S.S.P.A.
ARO4	Cipolla rossa di Acquaviva	40°54′3.348″ N 16°40′27.011″ E	Di.S.S.P.A.
ARO5	Cipolla rossa di Acquaviva	40°51′59.76″ N 16°53′0.527″ E	Di.S.S.P.A.
ARO6	Cipolla rossa di Acquaviva	40°52′48.72″ N 16°49′43.247″ E	Di.S.S.P.A.
ARO7	Cipolla rossa di Acquaviva	40°53′13.47″ N 16°50′23.783″ E	Di.S.S.P.A.
ARO8	Cipolla rossa di Acquaviva	40°53′18.816″ N 16°49′33.888″ E	Di.S.S.P.A.
ARO9	Cipolla rossa di Acquaviva	40°54′51.372″ N 16°49′3.504″ E	Di.S.S.P.A.
ARO10	Cipolla rossa di Acquaviva	40°54′1.188″ N 16°49′24.311″ E	Di.S.S.P.A.
ARO11	Cipolla rossa di Acquaviva	40°52′49.8″ N 16°49′48.575″ E	Di.S.S.P.A.
ARO12	Cipolla rossa di Acquaviva	40°52′38.892″ N 16°49′28.379″ E	Di.S.S.P.A.
ARO13	Cipolla rossa di Acquaviva	40°53′21.768″ N 16°49′29.711″ E	Di.S.S.P.A.
TRO1	Cipolla rossa lunga di Tropea	-	Di.S.S.P.A.
TRO2	Cipolla rossa tonda di Tropea	-	Di.S.S.P.A.
MCO	Cipolla ramata di Montoro	-	Di.S.S.P.A.

^†^ Di.S.S.P.A., Department of Soil, Plant and Food Sciences, University of Bari.

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
