# Peer review of "Assessment of Genetic Diversity of the “Acquaviva Red Onion” (Allium cepa L.) Apulian Landrace"

_plants, 2020, doi:10.3390/plants9020260_

Round 1

Reviewer 1 Report

The authors describe a study assessing the genetic diversity in onion landraces by means of 11 SSR markers. This research will contribute to the preservation of local landraces and limit commercial fraud. Both objectives are of high importance to diversity preservation, trade flows and economic prosperity, and consumers and farmers protection. Therefore, it is of high significance.

My comments will be organized from the less to the highest important:

Grammar and spelling

The article was carefully written and is of easy reading and understanding. Still, there are little typo mistakes that should be corrected to keep the quality of the manuscript:

Line

Found

Suggested

21, 177

A set of eleven.../... from the eleven

A set of 11... (as in line 114)

22 and others

... 13 Acquaviva red onion ...

... 13 ‘Acquaviva red onion’ (as it is in line 20)

25

... Montoro onions cvs...

... Montoro onions cultivars... (as in line 28)

28

... in the present study. Finally, the present study...

The repetition of “present study” sounds strange. Please, change one of them.

76

Draka et al., 2017

Dhaka et al., 2017 (as it is in “References”)

102

As reported in Table 1 the...

“As reported in Table 1, the...”

103, 174, 180

(ARO12)

(ARO 12)/ ARO 7 and ARO 12 / ARO 7 and ARO 12 (keep the space between the ARO and the numeral or remove it. The same for TRO)

103

...from 6.00... to 9.50 ...

°Brix is missing

106

...to 7.04 (???????)...

Please correct the (?????)

131

... chosen to genotyped our...

... chosen to genotype our...

139

...composed by twenty individuals ...

...composed by 20 individuals ...

Table 2

Ho = 0.3

Ho = 0.30

163

... subpopulation S4 dividing...

... subpopulation S5 standing out...

211

... 0.5 in fifty percentage...

... 0.5 in fifty percent ...

202, 257 others

Gonzalez-Perez et al.,

González-Pérez et al.,

235-238

... rest of the collection, probably...

... rest of the collection. Probably, ...

237

...visualize patter of genetic...

...visualize pattern of genetic... (??)

258

...material of twenty...

...material of 20 ...

259

... as A. cepa...

... as A. cepa...

288

... machine, which the alleles...

... machine, where the alleles...

Introduction

Line 60 - the reference used is quite old.

Line 88 - “…to assess genetic homogeneity…” - as the level of homogeneity is unknown, I would suggest to refer “genetic variability” instead.

Methodology

Lines 261 and 276 - The CTAB-based DNA extraction method extracts total DNA rather than only genomic DNA. I suggest substituting “genomic DNA” by “Total DNA”, as it is also used in line 262.

Line 272 - I suggest the authors to give an explanation for the choice of the SSR markers. For instance, ACM390 is described in the supplementary information of Baldwin et al., (2012), but was not used by these authors as is not among the 20 best ones.

Results

Line 180 - ARO is a landrace. I suggest the authors to replace “landraces” by “accessions”.

Discussion

Being ARO7 and ARO12 localizes as far from the group I as TROs and MCOs, can they be considered other landraces? Nothing is discussed about this observation.

Literature

Line 301 - Pritchard et al., 2000 is missing

Lines 306 and 307 - All references starting with “E”) are missing.

Line 369 - the year (2017) is missing.

Line 416 - Mallor et al (2014) is not in the text

Author Response

We acknowledge the reviewer for her/his suggestions and comments.

Point-to -point answers to his questions as reported below. English was thoroughly revised

Grammar and spelling

The article was carefully written and is of easy reading and understanding. Still, there are little typo mistakes that should be corrected to keep the quality of the manuscript:

Line

Found

Suggested

21, 177

A set of eleven.../... from the eleven

A set of 11... (as in line 114)

22 and others

... 13 Acquaviva red onion ...

... 13 ‘Acquaviva red onion’ (as it is in line 20)

25

... Montoro onions cvs...

... Montoro onions cultivars... (as in line 28)

28

... in the present study. Finally, the present study...

The repetition of “present study” sounds strange. Please, change one of them.

76

Draka et al., 2017

Dhaka et al., 2017 (as it is in “References”)

102

As reported in Table 1 the...

“As reported in Table 1, the...”

103, 174, 180

(ARO12)

(ARO 12)/ ARO 7 and ARO 12 / ARO 7 and ARO 12 (keep the space between the ARO and the numeral or remove it. The same for TRO)

103

...from 6.00... to 9.50 ...

°Brix is missing

106

...to 7.04 (???????)...

Please correct the (?????)

131

... chosen to genotyped our...

... chosen to genotype our...

139

...composed by twenty individuals ...

...composed by 20 individuals ...

Table 2

Ho = 0.3

Ho = 0.30

163

... subpopulation S4 dividing...

... subpopulation S5 standing out...

211

... 0.5 in fifty percentage...

... 0.5 in fifty percent ...

202, 257 others

Gonzalez-Perez et al.,

González-Pérez et al.,

235-238

... rest of the collection, probably...

... rest of the collection. Probably, ...

237

...visualize patter of genetic...

...visualize pattern of genetic... (??)

258

...material of twenty...

...material of 20 ...

259

... as A. cepa...

... as A. cepa...

288

... machine, which the alleles...

... machine, where the alleles...

A: all suggested changes were accepted 

Introduction

R: Line 60 - the reference used is quite old.

A: We changed the manuscript accordingly

R: Line 88 - “…to assess genetic homogeneity…” - as the level of homogeneity is unknown, I would suggest to refer “genetic variability” instead.

A: Suggestion accepted

Methodology

R: Lines 261 and 276 - The CTAB-based DNA extraction method extracts total DNA rather than only genomic DNA. I suggest substituting “genomic DNA” by “Total DNA”, as it is also used in line 262.

A: Suggestion accepted

R: Line 272 - I suggest the authors to give an explanation for the choice of the SSR markers. For instance, ACM390 is described in the supplementary information of Baldwin et al., (2012), but was not used by these authors as is not among the 20 best ones.

A: We choosed several SSRs available in literature, on the basis of PIC and map position. Due to the large samples number, at first we performed PCR to identify primer pairs that produced amplicon. Finally, we tested the 37 SSR primer pairs listed in Tab S3 on the 320 samples and only 11 of them provided scorable patterns. ACM390 did not produce scorable amplicons in our materials.

Results

R: Line 180 - ARO is a landrace. I suggest the authors to replace “landraces” by “accessions”.

A: We replaced the term “accession”, with “population” within the three landraces characterized in this study

Discussion

R: Being ARO7 and ARO12 localizes as far from the group I as TROs and MCOs, can they be considered other landraces? Nothing is discussed about this observation.

A:  ARO is a typical landrace cultivated in the area of Acquaviva, characterized by flat shape and red colour of the external skin. All the ARO populations characterized in this study (each one deriving from a different farmer which also multiply seeds),  display these general morphological features. Interestingly, our study shows that ARO7 and 12 are genetically different in spite of being phenotypically similar to other ARO populations. Probably, this s due to a different seed stock used by the two farmers. We now addressed this point in the discussion. 

Literature

R: Line 301 - Pritchard et al., 2000 is missing

A: Corrected.

R: Lines 306 and 307 - All references starting with “E”) are missing.

A: Corrected.

R: Line 369 - the year (2017) is missing.

A: Corrected

R: Line 416 - Mallor et al (2014) is not in the text

A: Corrected

Reviewer 2 Report

The manuscript entitled “Assessment of Genetic Diversity of ‘Acquaviva Red Onion’ (Allium cepa L.) Apulian Landrace” by Luigi Ricciardi et al characterized  use 11 SSR markers to estimate the genetic variation in 13 Acquaviva red onion accessions and 3 common commercial types. The genetic structure, hierarchical clustering  and principal coordinates  show the Acquaviva red onion represents a well-defined gene pool. The results presented in this manuscript are interesting and potentially significant, but some of the results of the study remain unclear or not sufficiently explored. Detail of comments are as outlined below.

line 106, there are some question maker need figure out. Line 318, should show the title of these supplementary materials. In table S3, need provide  the Annealing Temp. It will be impressive if the authors provide pictures of those onion phenotype,  the picture for checking the effectiveness on primer amplification and the PCR amplification products in onion varieties.

Author Response

We acknowledge the reviewer for her/his suggestions and comments.

Point-to -point answers to his questions as reported below. English was thoroughly revised

R: line 106, there are some question maker need figure out.

A: Corrected

R: Line 318, should show the title of these supplementary materials.

A: Corrected

R:In table S3, need provide  the Annealing Temp.

A: Done

R:It will be impressive if the authors provide pictures of those onion phenotype, the picture for checking the effectiveness on primer amplification and the PCR amplification products in onion varieties.

A: We added a picture of typical ARO bulbs (new Figure 1). Unfortunately SSR analyses were performed on a capillary sequencer that ,at the moment, is out of order, so we are not able to provide electropherograms

Reviewer 3 Report

The manuscript “Assessment of Genetic Diversity of ‘Acquaviva Red Onion’ (Allium cepa L.) Apulian Landrace” attempts to provide SSR molecular markers to assess the genetic diversity of ‘Acquaviva red onion’, and  discriminate this landrace from other onion cultivars. Overall, the manuscript conforms to the structure recommended by the journal viz. Introduction,
Materials and Methods, Results, and Discussion. Barring few typos, the overall English comprehension and grammar of the article is good and attains the standards of a scientific article. There is continuity in the text. Though I appreciate that authors have tried to conceptualize the observations in greater detail, I found details missing at a few places. Below are specific points that require improvements:  

-Line 28-29: Given that present study has not provided a novel mechanism, “Foundation” is a strong word in the light of present findings. I’d suggest using similar to “the present study is useful for…” or broaden our understanding …. ” etc. Please rephrase the sentence. 

-Typo in line 106 -At the beginning of section 2.2, it would be informative if the author could incorporate a few specific details, for e.g. about accessions/germplasm, the name of the average statistics calculated, etc.
-Based on the outcome in line 114-115, could the author comment on a few characteristics of those SSRs that showed polymorphism (n=11)? 

-In the light of observation (line 180-181, figure 2a,b), it would be interesting and useful if the author could discuss/comment/show on possibilities/mechanisms/characteristics that could be attributed to distinguish ARO12 and ARO7 from the rest of the AROs. 

-With reference to Figure 2, it would be useful if authors can additionally provide the raw file of the genetic distance matrix as a supplementary data. That would be important for the reproducibility of the results.

Author Response

We acknowledge the reviewer for her/his suggestions and comments.

Point-to -point answers to his questions as reported below. English was thoroughly revised

R:Line 28-29: Given that present study has not provided a novel mechanism, “Foundation” is a strong word in the light of present findings. I’d suggest using similar to “the present study is useful for…” or broaden our understanding …. ” etc. Please rephrase the sentence. 

A: The sentence has been rephrased accordingly

R:-Typo in line 106 –

A: Corrected

R: At the beginning of section 2.2, it would be informative if the author could incorporate a few specific details, for e.g. about accessions/germplasm, the name of the average statistics calculated, etc.

A: As suggested by Reviewer 2, we added a figure in which bulbs of Acquaviva red onion are shown. Regarding average statistics calculated, they are described in Table 2 and in the Material and method section. You might specify if other details are needed.

R:-Based on the outcome in line 114-115, could the author comment on a few characteristics of those SSRs that showed polymorphism (n=11)? 

A: We choosed several SSRs available in literature, on the basis of PIC and map position. Due to the large number of samples, at first, we performed PCR to identify primer pairs that produced amplicons. Finally, we tested the 37 SSR primer pairs listed in Tab S3 on the 320 samples and only 11 of them provided scorable patterns

R:-In the light of observation (line 180-181, figure 2a,b), it would be interesting and useful if the author could discuss/comment/show on possibilities/mechanisms/characteristics that could be attributed to distinguish ARO12 and ARO7 from the rest of the AROs. 

A: ARO is a typical landrace cultivated in the area of Acquaviva, characterized by flat shape and red colour of the external skin. All the ARO populations characterized in this study (each population referring to an individual farm) display these general morphological features. Interestingly, our study shows that ARO7 and 12 are genetically different in spite of being phenotypically similar to other ARO populations. Probably, genetic peculiarity of ARO7 and ARO12 is due to different seed stocks used by the two farmers.  Further phenotypic characterization might reveal peculiar phenotypic traits associated with these two populations.

R:-With reference to Figure 2, it would be useful if authors can additionally provide the raw file of the genetic distance matrix as a supplementary data. That would be important for the reproducibility of the results.

A: The matrix was  added in the supplementary material

Reviewer 4 Report

The current article (Assessment of Genetic Diversity of ‘Acquaviva Red Onion’ (Allium cepa L.) Apulian Landrace) aims to study the genetic diversity and the preservation of onion landraces cultivated locally in Italy. This work is very relevant considering the importance of landraces for their adaptation to environment and being an backup for commercial cultivars.

Although this article describe clearly the aims and the results of the study, I have some remarks:

The CV values are very high mainly for pungency, any explanation? Line 106 needs to be reviewed (remove ?? and replace it with ARO 7) Line 114, the author says (11 out of 37 tested SSR primer combinations provided single-locus polymorphisms), how do you know it is a single locus? What is the maximum number of allele found in a single individual? Analysis of molecular variance (AMOVA) showed that the variation within accessions is significantly high. This was done using the data of marker scoring of the 320 studied individuals? The input data for AMOVA and for STURCTURE software is not clear enough. Do you have missing values, and if yes how did you deal with it? Any marker had a miner allele frequency? For STURCTURE software and for PCoA analysis need to be performed on the level of individuals not on the level of accessions considering the high and significant genetic diversity within accession. ARO7 is very distinct form other accessions (the first dimension of PCoA is very high 73.6%), is there any explanation? Geographically? Is it crossable with other accessions? The study will be more valuable when wild species is used as an outgroup for the dendrogram analysis.

Author Response

We acknowledge the reviewer for her/his suggestions and comments.

Point-to -point answers to his questions as reported below. English was thoroughly revised

R:The CV values are very high mainly for pungency, any explanation?

A: As revealed by molecular analyses, some AROs showed  a wide intra- population genetic diversity,  we think this variabilty may also affect several traits, such as pungency.

R: Line 106 needs to be reviewed (remove ?? and replace it with ARO 7)

A: Corrected

R: Line 114, the author says (11 out of 37 tested SSR primer combinations provided single-locus polymorphisms), how do you know it is a single locus? What is the maximum number of allele found in a single individual?

A: The 11 SSR markers detected single locus polymorphism in our genotypes, meaning that in a single individual we detect at most 2 alleles. We specified this issue in the text.

R: Analysis of molecular variance (AMOVA) showed that the variation within accessions is significantly high. This was done using the data of marker scoring of the 320 studied individuals? The input data for AMOVA and for STURCTURE software is not clear enough. Do you have missing values, and if yes how did you deal with it? Any marker had a miner allele frequency?

A: We performed Analysis of Molecular Variance by means of GenAlEx software. As supposed, the analysis was computed using our codominant data, scored for each of the 320 individuals, as input data set. The same data set was also used as input file for population structure analysis implemented in STRUCTURE software. A few missing values were present in our data set and they were properly coded as required for both the software. 

R: For STURCTURE software and for PCoA analysis need to be performed on the level of individuals not on the level of accessions considering the high and significant genetic diversity within accession.

The study will be more valuable when wild species is used as an outgroup for the dendrogram analysis.

A: In this research, for the first time, we investigated the genetic variation of an Apulian red onion landrace very important for the income of the local farmers, as well as representing an excellent Apulian food product exported both to several Italian regions and to other countries. We wanted to assess the genetic profile of this Apulian landrace and, in order to protect this niche product, we particularly wanted to compare Acquaviva red onion landrace with other two more famous red onion cvs. We did not stress the high level of genetic variation obtained within accession, since it is expected for outcrossing crops and in random mating condition. Remarkably, the study showed as Acquaviva red onion landrace forms a different genetic cluster from Tropea and Montoro onions. We emphasized this result because Tropea and, specially, Montoro onions are very often mistaken with Acquaviva red onion causing commercial frauds frustrating the valuable endeavour of the local farmers. 

R: ARO7 is very distinct form other accessions (the first dimension of PCoA is very high 73.6%), is there any explanation? Geographically? Is it crossable with other accessions?

A: ARO is a landrace cultivated in the town of Acquaviva characterized by flat shape and red colour of the external skin. All the ARO populations genotyped in this study (each population derived from a single farm), display these typical phenotypic traits. ARO7 is crossable with the other populations, we think that its diversity is due to the use of different seed stock by the farm in which it was collected. We now addressed this point in the discussion.

Round 2

Reviewer 2 Report

Accept in present form.

Author Response

Dear Reviewer,

thank you very much for your comment. As you suggested, we reviewed the english style